# Effect of High $CO_2$ Treatment and MA Packaging on Sensory Quality and Physiological-Biochemical Characteristics of Green Asparagus (*Asparagus officinalis* L.) during Postharvest Storage

**Li-Xia Wang** [1,†], **In-Lee Choi** [1,2,†] **and Ho-Min Kang** [1,3,*]

1    Division of Horticulture and Systems Engineering, Kangwon National University, Chuncheon 24341, Korea; 2018lisa@kangwon.ac.kr (L.-X.W.); cil1012@kangwon.ac.kr (I.-L.C.)
2    Agricultural and Life Science Research Institute, Kangwon National University, Chuncheon 24341, Korea
3    Interdisciplinary Program in Smart Agriculture, Kangwon National University, Chuncheon 24341, Korea
*    Correspondence: hominkang@kangwon.ac.kr; Tel.: +82-33-250-6425; Fax: +82-33-259-5562
†    Li-Xia Wang and In-Lee Choi contributed equally to this work.

**Abstract:** Green asparagus is vulnerable to thrips that carry microorganisms and cause deterioration in quality. The effects of 60% $CO_2$ treatment, which is used to kill thrips, combined with perforated (P) or modified atmosphere (MA) packages during cold storage, on the sensory quality and physiological–biochemical characteristics of asparagus were investigated. MA packaging yielded an asparagus shelf-life five days longer than P packaging. The 60% $CO_2$ treatment for 48 h at 4 °C packaged with MA film ($CO_2$-48 h-4 °C-MA) showed a lower number of aerobic bacteria, yeast, and mold. Yellowing of asparagus was retarded, as shown by higher hue angle and chlorophyll content and lower chlorophyllase activity. Also, $CO_2$-48 h-4 °C-MA treatment inhibited the reduction of soluble solids content in asparagus. Likewise, all high $CO_2$ treatments showed lower electrolyte leakage (EL), with $CO_2$-48 h-4 °C-MA demonstrating the minimum EL. The effectiveness of high $CO_2$ on maintaining sensory qualities was observed, with a score higher than 3.0. In conclusion, $CO_2$-48 h-4 °C-MA treatment during cold storage was effective for maintaining post-harvest sensory qualities and physiological–biological traits of asparagus, and provided strong inhibition of microflora growth during the storage period.

**Keywords:** green asparagus; high $CO_2$; modified atmosphere packages; sensory quality; physiological and biochemical characteristics; postharvest cold storage

## 1. Introduction

Asparagus (*Asparagus officinalis* L.) is an herbaceous perennial plant that is widely cultivated and harvested around the world. Green asparagus is considered a health food, with fresh or fresh-cut asparagus becoming increasingly popular due to its structural and sensory qualities and abundant nutritional components [1], such as vitamins, minerals, and dietary fiber [2]. Asparagus has a short shelf-life due to its high respiratory rate of 60 mg $CO_2$/kg/h at 5 °C, metabolic processes, and pathogenic microorganisms that lead to decay and quality deterioration during postharvest storage [3]. Thrips not only cause damage, such as collapse and soft rot on flowers and vegetables, but also transmit viruses [4] and diseases. The effect of high levels of carbon dioxide ($CO_2$) on killing insect pests in agricultural products is attracting great attention, and is widely used in many countries on many kinds of vegetables. Increased $CO_2$ is likely to affect the biology of insect herbivores and alter the leaf damage caused by these pests [5]. A previous study with strawberry confirmed that elevated carbon dioxide or

oxygen strengthened multiple defense pathways by increasing resistance against bacterial and fungal attacks [6,7]. High $CO_2$ content, which causes 100% lethality of adult onion thrips within 24 h of treatment, was shown to be an effective insecticide for the adults and larvae of onion thrips [8]. Treatment with 60% $CO_2$ for 12 h achieved 100% lethality in most thrips [9].

As a substrate of photosynthesis and the product of respiration, $CO_2$ is important for plants both in the field and during storage. High $CO_2$ influences the quality of strawberries by affecting respiration and secondary metabolism [10]. Elevated $CO_2$ atmosphere was shown to extend the shelf-life of horticultural products, inhibit browning [11], restrain decay incidence caused by the development of microorganisms [12], and suppress activities of cellulase, polygalacturonase, and peroxidase [13]. Blanch [14] reported that short-term, high-$CO_2$ treatment maintained high sucrose levels and mitigated certain physiological and structural disorders.

A sharp decrease in chlorophyll content and the green color of asparagus were reported after eight days of storage [15]. The degradation of chlorophyll was regulated by a multistep pathway, with chlorophyllase one of the key enzymes potentially induced by exogenous stimuli [16]. Different effects of high-$CO_2$ treatments on chlorophyll were reported. High $CO_2$ maintained high chlorophyll levels until the end of the storage period in table grapes [17] and led to stable color properties (*L\**, *a\**, *C\**, *h°*) in strawberry [18]. A rapid decrease in citrus chlorophyll was induced by elevated $CO_2$, with simultaneous upregulation of the related genes *itSGR*, *CitNYC*, *CitChlase*, *CitPPH*, *CitPaO,* and *CitRCC* [19]. A short-term, high-$CO_2$ treatment at low temperatures maintained quality and prolonged the shelf-life of grapes [20]. In this study, we investigated the effects of high-$CO_2$ treatment with different durations at ambient and cold temperatures on the sensory qualities and physiological–biochemical characteristics of green asparagus during cold storage.

## 2. Materials and Methods

### 2.1. Plant Materials

Green asparagus (*Asparagus officinalis* L. cv. "Welcome") was cultivated on a farm located in Yanggu-gun, Gangwon-do, Korea. Asparagus was harvested on the same day in October 2018. After harvest, asparagus was immediately transported to the Postharvest Physiology and Distribution Laboratory of Kangwon National University. Asparagus with a diameter of 1.4 ± 0.1 cm and length of 24 ± 1.0 cm were selected and randomized for this study. All asparagus were placed in a 4 °C refrigerator before being treated.

### 2.2. Packaging Materials, Storage Conditions, and Replication

Modified atmosphere (MA) packages with a 10,000 cc/m²·day·atm oxygen transmission rate (OTR) (Dae Ryung Precision Packaging Industry Co., Ltd., Gwangju-si, Korea) and perforated (P) packages (10,000 cc/m²·day·atm film with 4 holes of 0.5 cm diameter) were used to package the asparagus.

$CO_2$ treatments were conducted at 20 °C for 24 h and at 4 °C for 48 h and 72 h. $CO_2$ (released from dry ice blocks) was applied in a plastic box (30 × 21 × 15 cm) at 60% (*v/v*), and the content of $CO_2$ was evaluated every 3 h (see below). Untreated asparagus placed in ambient air (20 °C) for 24 h or at 4 °C in a refrigerator for 60 h served as controls. Afterward, all asparagus was packaged with MA or P packages. All treatment combinations are shown in Table 1.

**Table 1.** Treatment conditions.

| Treatments | | 24 h-20 °C | 48 h-4 °C | 72 h-4 °C |
|---|---|---|---|---|
| Air | P [z] | Air-24 h-20 °C-P | Air-60 h-4 °C-P | |
| | MA | Air-24 h-20 °C-MA | Air-60 h-4 °C-MA | |
| $CO_2$ | P | $CO_2$-24 h-20 °C-P | $CO_2$-48 h-4 °C-P | $CO_2$-72 h-4 °C-P |
| | MA | $CO_2$-24 h-20 °C-MA | $CO_2$-48 h-4 °C-MA | $CO_2$-72 h-4 °C-MA |

[z] P: Perforated packages (10,000 cc/m²·day·atm film with 4 holes of 0.5 cm diameter); MA: Modified atmosphere packages with a 10,000 cc/m²·day·atm oxygen transmission rate.

Each treatment was replicated 5 times, and the package unit consisted of three asparagus spears with a total fresh weight of 65–75 g. All treatments were stored at 4 ± 1 °C, with a relative humidity of 85 ± 5%.

## 2.3. Microbiological Analysis

Microorganisms on the asparagus were measured on the initial and final storage days. Approximately 2.0 g of asparagus tissue was mixed with 18 mL diluent (sterilized water) using a stomacher (Powermixer, B&F Korea, Gimpo-si, Korea) set at the highest speed (level 10, 200 rpm) for 3 min. Then, the mixture was diluted 1000-fold. The dilution (1 mL) was dropped onto 3 M microbiology count plate petrifilms (3M Co., St. Paul, MN, USA). Then, aerobic bacteria was cultivated at 35 °C for 72 h, and yeasts and molds were cultivated at 25 °C for 72 h. According to Wang [21], the growth of total aerobic bacteria, yeasts, and molds, were measured using the Petrifilm Plate Reader (3M Co., St. Paul, MN, USA). The concentrations of microorganisms were reported in log colony-forming units $(CFU) \cdot g^{-1}$.

## 2.4. Sensory Quality

The fresh weight-loss rate was measured according to the following formula:

$$\text{Weight loss}(\%) = \frac{\text{Initial fresh weight} - \text{Final fresh weight}}{\text{Initial fresh weight}} \times 100\% \tag{1}$$

Five skilled and experienced panelists from the Postharvest Physiology and Distribution Laboratory assessed the general acceptability of the asparagus following the criteria of Wang [21]. Visual quality and off-odor on the final storage day were determined. Asparagus visual quality was assessed on a scale of 1 to 5 (1 = worst, 5 = best), and off-odor was assessed on a scale of 0 to 5 (0 = no off-odor and 5 = strong off-odor). Asparagus with a visual quality score equal to or greater than 3 and an off-odor score of less than 3 were determined to be suitable for sale.

## 2.5. Atmosphere Analysis

The headspace carbon dioxide ($CO_2$) and oxygen ($O_2$) content in the different treatments were measured with an infrared $CO_2/O_2$ analyzer (Model Check Mate 9900, PBI-Dansensor, Ringsted, Denmark) by pressing the needle of the measuring assembly through a septum adhered to the packaging film, with the measurement starting automatically. The ethylene ($C_2H_4$) content was measured with a GC-2010 Shimadzu gas chromatograph (GC-2010, Shimadzu Corporation, Japan) equipped with a BP 20 Wax column (30 m × 0.25 mm × 0.25 um, SGE analytical science, Australia) and a flame ionization detector (FID). The detector and injector operated at 127 °C and the ovens at 50 °C. The carrier gas ($N_2$) flow rate was 0.67 $mL \cdot s^{-1}$, with 1.0 mL gas extracted from packages and inserted into the GC-2010 Shimadzu gas chromatograph.

## 2.6. Physicochemical and Biochemical Traits

The firmness of the asparagus tips (from the topmost 5 cm) and stems (from lowermost 8 cm) was measured with a rheometer (Compac-100, Sun Scientific Co., Ltd., Tokyo, Japan) using a probe (Ø 3.0 mm) at 1.0 mm/s speed.

The hue angle of the asparagus spear tips and stems was measured using a color-difference meter (Model CR-400, Konica Minolta Sensing, Inc., Osaka, Japan).

Soluble solids content (SSC) was measured using a pocket refractometer (PAL-1, Atago, Tokyo, Japan). Asparagus samples were chopped up and extruded with gauze wrapping. The solution obtained from the asparagus was directly dripped onto a pocket refractometer and the results were read directly in °Brix.

Electrolyte leakage (EL) was measured following the protocol described by Wang [21]. Pieces of asparagus samples (0.6 g) were submerged in 0.4 M mannitol (25 mL) for 3 h at ambient temperature (20 ± 1 °C), then the EL was immediately measured using a handheld meter (HI 9813-6 Portable pH/EC/TDS/ °C Meter; Hanna, Instruments, Padova, Italy). Then, samples were frozen at −20 °C and thawed at ambient temperature; this was repeated, and the solution was measured three times. The EL was calculated as follows:

$$\text{Relative electrolyte leakage (EL)} = \frac{\text{EL}_i}{\text{EL}_f} \times 100\% \tag{2}$$

To calculate total chlorophyll content following Yoon [22], with slight modification, a 1.0 g piece of tip or stem tissue was chopped up and dissolved in 10 mL of methanol and held in the dark at 4 °C for 48 h to extract chlorophyll. Absorbance values at 642.5 nm (A642.5) and 660 nm (A660) were measured using a UV-VIS spectrophotometer (UV mini model 1240, Shimadzu, Tokyo, Japan). Total chlorophyll content was calculated using the following formula:

$$\text{Total chlorophyll} \left(\text{mg·mL}^{-1}\right) = 7.12 \times A_{660} + 16.8 \times A_{642.5} \tag{3}$$

The chlorophyllase enzyme [EC 3.1.1.14] was extracted from the asparagus tip and stem tissue, and chlorophyllase activity was determined by measuring the formation of chlorophyllides [23]. A 2.0 g piece of frozen asparagus tissue was blended in a mixer with cold acetone and immediately washed three times with cold acetone. The pigmented residue was washed with cold acetone until the powder was colorless, and the enzymatic acetone extract was obtained by homogenizing powder in 3 mL of sodium phosphate buffer (5 mM) at a pH of 7.0, containing Triton X-100 (0.24%, *w/v*), polyvinyl pyrrolidone (2.5%, *w/v*) and potassium chloride (50 mM). The homogenate was centrifuged at 12,000× *g* for 10 min at 4 °C, where the supernatant was the crude enzyme extract. Chlorophyllase activity was determined using a reaction mixture of 30 uL 0.1 M ascorbic acid, 1.7 mL acetone, 0.3 M 6.8 mg·mL$^{-1}$ chlorophyll, and 0.5 mL crude enzyme, adding distilled water for a 3.3 mL total volume. After 40 min in the dark at 35 °C, 3 mL of acetone was added to stop the reaction, then 6.0 mL of hexane and 1.0 mL of 10 mM KOH were added to separate the chlorophyllide. The homogenate was centrifuged at 12,000× *g* for 10 min at 4 °C, with the lower phase containing the chlorophyllide a. The reaction mixture (enzyme-free) was used as a blank. Absorption was read at 667 nm to measure chlorophyllase activity.

### 2.7. Statistical Analysis

The experiment was a completely randomized design with five replications of each treatment combination. Microsoft Excel 2016 and IBM SPSS Statistics (24, IBM Corp., Armonk, NY, USA) were used to statistically analyze the data. Significant differences were tested using ANOVA (one-way analysis of variance) and means were compared using the least significant difference (LSD) test at $p < 0.05$.

## 3. Results and Discussion

### 3.1. Atmosphere Analysis in All Packages

The atmospheric conditions in packages affect fresh produce qualities, such as flavor, color, firmness, sugar, pH, and acidity [24]. A modified atmosphere (MA) package modifies the gas environment in packages by allowing oxygen transmission through the film. The content of $O_2$ in packages reached equilibrium after one day of storage and stabilized at nearly 21% in P packages and 19% in MA packages (Figure 1A,B). Atmospheric composition in MA packages was modified during the storage period, and no significant differences among prepackaging treatments were observed within the same package type. The atmospheric composition determines the respiration rate of the harvested produce, thereby causing deterioration [25]. The content of $O_2$ in packages depends only on packaging film,

signifying that high-$CO_2$ treatments cause no irreparable damage to asparagus. An $O_2$ content higher than 5% is better for asparagus cold storage [26]. Likewise, an $O_2$ below 5–10% is harmful to green asparagus and leads to an accumulation of ethanol, which can cause alcoholic off-flavors [27].

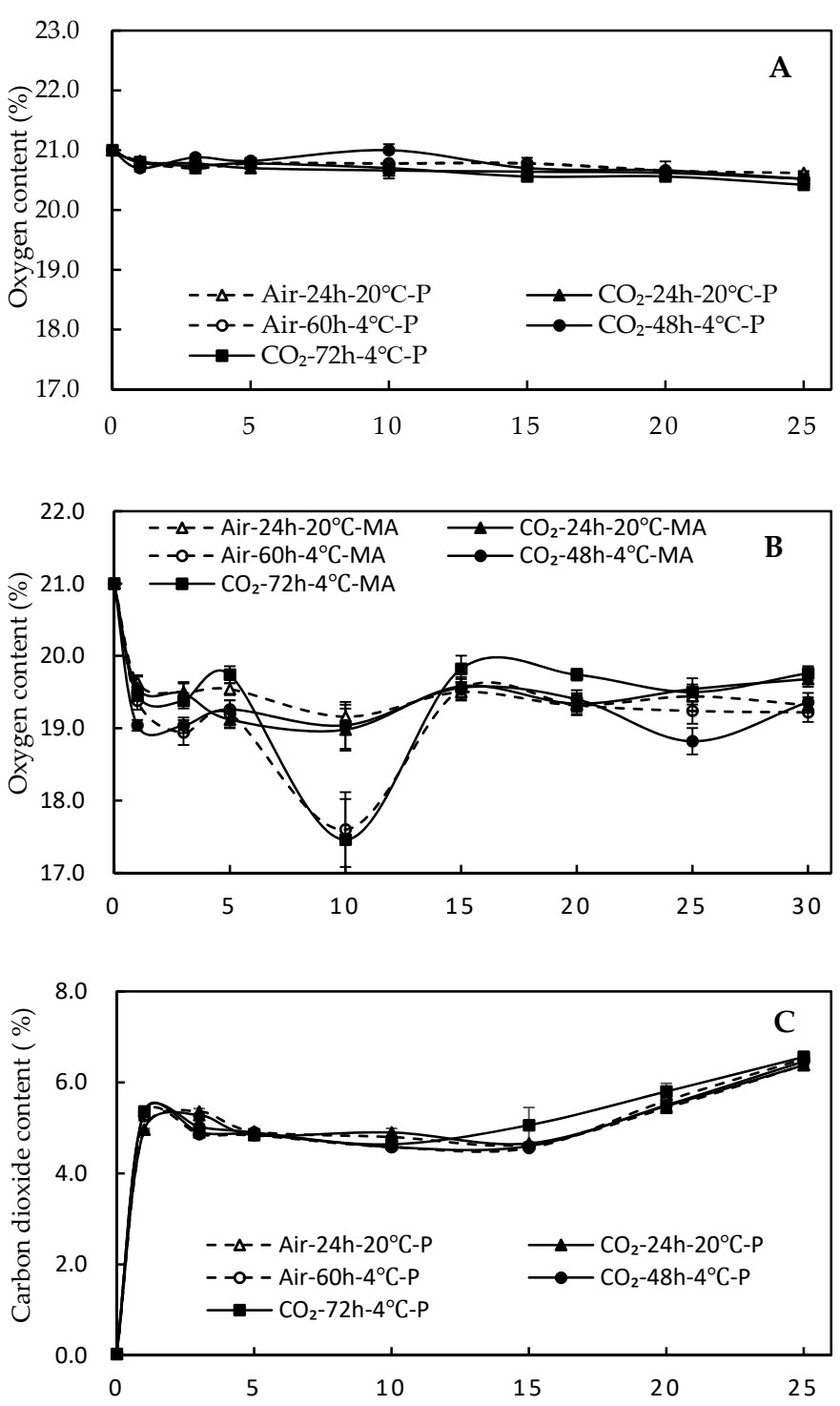

**Figure 1.** *Cont.*

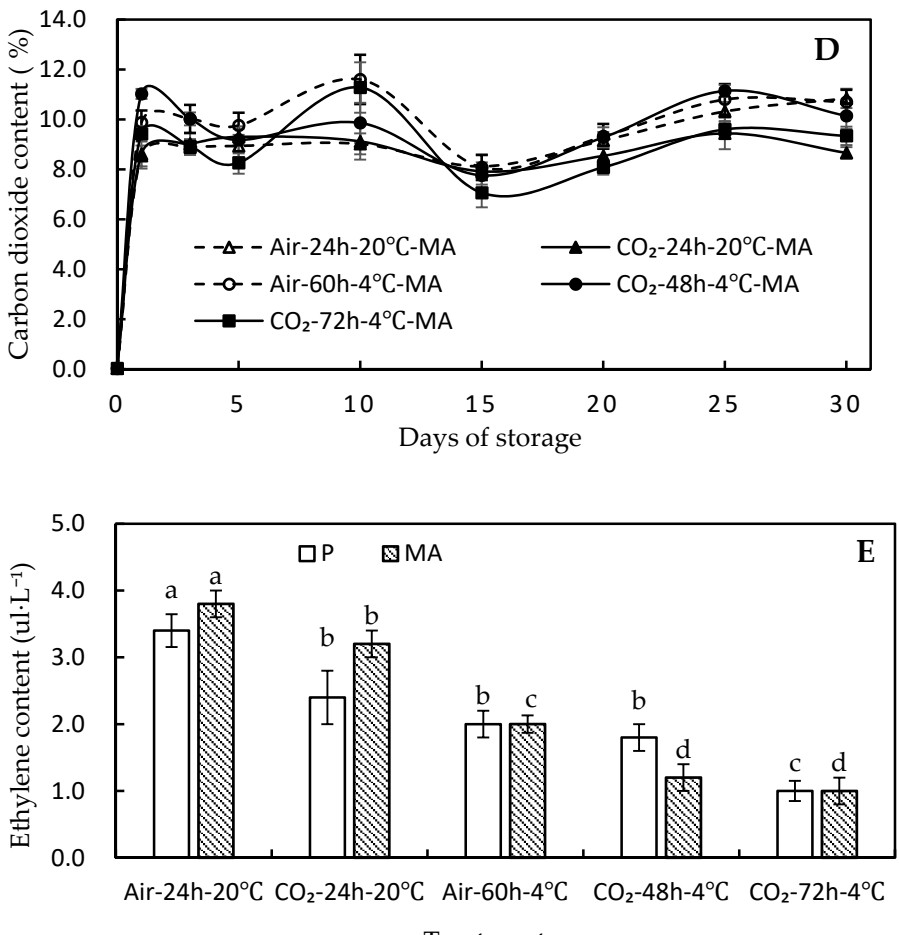

**Figure 1.** Changes in oxygen (**A**,**B**), carbon dioxide (**C**,**D**), and ethylene (**E**) contents in perforated (P) and modified atmosphere (MA) packages. (**A**,**C**): Perforated packages; (**B**,**D**): 10,000 cc/m$^2$·day·atm oxygen transmission rate (OTR)-modified atmosphere packages (MA). Treatments were 60% $CO_2$ or air for 24 h at 20 °C, 60% $CO_2$ for 48 or 72 h, or air for 60 h at 4 °C. All the groups were packaged and stored at 4 °C until the final day. Values are presented as means ± SE (*n* = 5). Different letters among the same packaging types indicate a statistical difference using an LSD (Least Significant Difference) test at *p* < 0.05.

$CO_2$ content rapidly increased on the first storage day and then remained stable (Figure 1C,D). The content of $CO_2$ was nearly 10% in the MA packages and nearly 6% in the P packages. Similar to the changes in oxygen, the content of $CO_2$ in the MA packages fluctuated slightly more than in the P packages during storage. Similar contents of $CO_2$ in all high-$CO_2$ treatments packaged with the same film were observed. The different temperature treatments showed no significant effects on the content of $CO_2$ in packages during cold storage. All the treatments maintained a suitable content of $CO_2$ for asparagus storage, 5–12% [26]. A $CO_2$ content above 15% was suggested to be harmful to green asparagus at optimal temperature [27], but high-$CO_2$ treatment showed no negative effect on the $CO_2$ content in asparagus storage packages.

Ethylene contents in all packages were measured on the 20th day. In this study, regardless of the package type, the content of ethylene was lower in high-$CO_2$ treatments, and as the treatment time increased, the inhibition increased (Figure 1E). Ethylene can induce product damage and lead to some undesirable effects by accelerating senescence [24,28]. It was previously reported that the content of ethylene increased after asparagus harvest until the final day of storage. As a non-climacteric vegetable, asparagus exhibited a high respiration rate and metabolic activity, which is related to

the regulation of ethylene [29]. $CO_2$ is a competitive inhibitor of ethylene; high $CO_2$ inhibits the expression of *NuERF2/5* and induces *ERF1* and *ERF4,* which was shown to correlate highly with the degree of browning in lotus [30]. Fortunately, the content of ethylene was lower than 4.0 uL·$L^{-1}$ in air treatments and lower than 2.0 μL·$L^{-1}$ in high-$CO_2$ treatments. The content of ethylene in air treatments probably reflects advanced deterioration after harvest, and the accumulation of $CO_2$ in packages affects ethylene production [31]. Cutting or wounding was shown to induce the production of ethylene (1.9 μL·$L^{-1}$) in asparagus [27], as similarly shown in our data (2.0 μL·$L^{-1}$), due to activation of 1-aminocyclopropane-1-carboxylic acid (ACC) synthase and ACC oxidase [32].

### 3.2. Effect of High-$CO_2$ Treatments on Physiochemical Traits

Texture, soluble solids content (SSC) and color are the critical factors in the acceptability of fruits and vegetables by consumers. The texture of asparagus, which is one of the most important qualities [33], relates to firmness. In this study, firmness was measured at the tip and stem before treatment and on the last storage day. Compared with the initial day, firmer asparagus was detected on the final day of storage, indicating an increase in firmness during storage (Table 2). The $CO_2$-72 h-4 °C and Air-60 h-4 °C treatments yielded the lowest firmness values for tips and stems packaged with P. Regardless of the packages, in all treatments the firmness of the tip was greater than in the stem. No notable difference was found in the tips packaged with MA, while a significant difference was observed between stems. High-$CO_2$ treatment slightly increased the stem firmness of asparagus, but not the tip. Firmness of the asparagus is a symptom of senescence and is related to the lignification of the fiber rings and vascular bundles. It was influenced by high temperature. These results were consistent with research showing that reducing the temperature could maintain the asparagus textural quality by controlling enzymes such as phenylalanine ammonia-lyase and peroxidase [27]. Regardless of the storage temperature, less toughness was observed in $CO_2$-treated asparagus after 15 days of storage compared with untreated control samples [34].

**Table 2.** Changes in firmness and soluble solids content (SSC) in green asparagus tip and stem tissues packaged with P or MA films on the initial and final storage days (the 25th day in P packages and the 30th day in MA packages).

| Treatments | Firmness (N) | | | | SSC (°Brix) | |
| --- | --- | --- | --- | --- | --- | --- |
| | Tip | | Stem | | | |
| | P | MA | P | MA | P | MA |
| Initial day | 7.52 b [z] | 7.96 b | 7.52 b | 7.96 a | 4.75 a | 4.75 a |
| Air-24 h-20 °C | 11.02 a | 12.08 a | 7.76 ab | 8.50 a | 4.11 b | 3.40 c |
| $CO_2$-24 h-20 °C | 10.30 a | 10.80 a | 8.82 a | 7.26 a | 4.17 b | 3.13 d |
| Air-60 h-4 °C | 10.18 a | 11.70 a | 7.32 b | 8.96 a | 4.30 b | 3.68 bc |
| $CO_2$-48 h-4 °C | 10.56 a | 12.22 a | 8.90 a | 8.02 a | 4.30 b | 3.78 b |
| $CO_2$-72 h-4 °C | 10.04 a | 11.92 a | 8.28 ab | 8.02 a | 4.09 b | 3.56 c |

[z] Values are presented as means ± SE (*n* = 5). Different letters in the same column indicate that the means are statistically different using an LSD (Least Significant Difference) test at *p* < 0.05.

Compared to the initial day, soluble solids content (SSC) decreased in all treatments. Green asparagus in MA packages exhibited lower soluble solids contents than those in P packages on the final storage day. The lowest soluble solids content was observed in the $CO_2$-24 h-20 °C-MA treatment, probably due to higher temperature. No obvious difference was found among treatments in P packages, whereas an obvious difference was observed in MA packages (Table 2), consistent with studies showing that a higher content of $CO_2$ for longer periods of time resulted in damaged asparagus [35].

Yellowing is a visual indicator of deterioration during long-term storage. To determine the degree of yellowing in asparagus, hue angle (0° = red-purple; 180° = bluish-green) and chlorophyll content were measured. Compared to the initial day, the hue angle of both tips and stems in the

$CO_2$-48 h-4 °C-MA treatment showed a minimal decrease and was closest to 120, the value for green. In both package types, the $CO_2$-24 h treatment showed the greatest amount of yellowing, in agreement with a previous report stating that hue angle remained high in the high-$CO_2$ treatment [36]. High $CO_2$ was verified to inhibit the yellowing of asparagus based on larger hue angle values (Table 3). On the initial day, the contents of chlorophyll were as high as 9.97 mg·mL$^{-1}$ and 8.33 mg·mL$^{-1}$ in the tip and stem, respectively, and the content significantly decreased by the last storage day. Regardless of the package type, $CO_2$-48 h and Air-60 h treatments produced higher contents of chlorophyll than 5.0 mg·mL$^{-1}$ on the last storage day. The lowest chlorophyll content was observed with $CO_2$-24 h-20 °C, which was less than 4.0 mg·mL$^{-1}$, implying that cold temperature and a suitable duration of high $CO_2$ treatment inhibited the degradation of chlorophyll. High temperature is more likely to lead to asparagus yellowing and chlorophyll degradation. Elevated $CO_2$ retarded the yellowing at low temperature due to the degradation of chlorophyll, as pointed out with cucumber and broccoli [37], which were shown to be similar in terms of change in hue angle as this study. Color change with $CO_2$-48 h was slightly more than in the control and $CO_2$-72 h treatment groups. There were considerable treatment-specific differences in hue angle changes and chlorophyll content. The breakdown of chlorophyll is a multistep pathway regulated by key enzymes such as chlorophyllase, Mg-dechelatase, pheophorbidea oxygenase, and red chlorophyll catabolite reductase [38]. Chlorophyllase is the first enzyme in the pheophorbide an oxygenase (PAO) pathway of chlorophyll degradation. The formation of chlorophyllides was used to evaluate the chlorophyllase activity in this study. In green asparagus, an inverse relationship was shown between chlorophyllase activity and the content of chlorophyll. The lowest activity in the tip (4.35 U·g$^{-1}$) and stem (3.36 U·g$^{-1}$) were observed on the initial day, and Air-24 h and $CO_2$-24 h showed higher enzyme activity levels, nearly 20.0 U·g$^{-1}$ (Table 3), consistent with Chinese flowering cabbages (*Brassica. rapa* var. parachinensis), in which enzyme activity peaked on the second day of storage [38]. Ramin and Modares [39] confirmed that increased $CO_2$ content was useful for maintaining the color of green olives. The content of chlorophyll is regulated by many related genes. High-$CO_2$ treatment in Chinese cabbage decreased the activity of chlorophyllase and lowered transcript abundances of the *BrChlase*, *BrPAO*, and *BrRCCP* genes, which were shown to regulate key enzymes in chlorophyll degradation after four days of storage [38]. In contrast, a decrease in *CitCAB* transcript levels and increased *CitSGR* transcript levels under elevated $CO_2$ was observed, which may contribute to chlorophyll degradation [38]. *itCAB* and *CitSGR* expression are related to chlorophyll state [19]. In cherry tomato (*Solanum lycopersicum* var. "Unicon"), the application of 5% $CO_2$ was better than air in maintaining high-quality skin color, ethylene production rate, and respiration rate during cold storage [40]. These very different results are probably influenced by the content of $CO_2$, plant species, and storage conditions [19].

**Table 3.** Hue angle, chlorophyll content, and chlorophyllase activity of green asparagus in P (the 25th day) and MA (the 30th day) packages.

| Treatments | Hue Angle (∘) | | | | Chlorophyll (mg·mL$^{-1}$) | | | | Chlorophyllase Activity (U·g$^{-1}$) | | | |
|---|---|---|---|---|---|---|---|---|---|---|---|---|
| | Tip | | Stem | | Tip | | Stem | | Tip | | Stem | |
| | P | MA | P | MA | P | MA | P | MA | P | MA | P | MA |
| Initial day | 125 a z | 125 a | 122 a | 122 a | 10.0 a | 10.0 a | 8.3 a | 8.3 a | 4.4 c | 4.4 e | 3.4 c | 3.4 c |
| Air-24 h-20 °C | 120 c | 122 b | 119 cd | 121 b | 4.5 c | 4.7 c | 4.7 c | 4.2 d | 20.7 a | 14.4 bc | 8.2 bc | 9.4 bc |
| $CO_2$-24 h-20 °C | 120 c | 112 c | 118 d | 118 c | 3.9 e | 4.0 e | 4.2 d | 4.2 d | 14.3 b | 12.5 cd | 17.6 a | 11.7 b |
| Air-60 h-4 °C | 121 bc | 122 b | 121 b | 119 b | 5.0 b | 5.6 b | 4.8 c | 5.3 c | 20.1 a | 21.8 a | 13.6 ab | 11.4 b |
| $CO_2$-48 h-4 °C | 122 b | 122 b | 121 b | 121 b | 5.2 b | 5.7 b | 5.4 b | 5.7 b | 13.1 b | 10.3 d | 15.1 a | 13.6 ab |
| $CO_2$-72 h-4 °C | 120 c | 122 b | 119 c | 119 b | 4.1 d | 4.5 d | 4.2 d | 4.3 d | 14.2 b | 16.2 b | 15.8 a | 18.8 a |

z Values are presented as means ± SE (*n* = 5). Different letters in the same column indicate that the means are statistically different using an LSD (Least Significant Difference) test at *p* < 0.05.

### 3.3. Effect of High-$CO_2$ Treatments on Biochemical Traits

Electrolyte leakage, which could reflect the integrity of the microsomal membrane, is widely used to evaluate injury to plant tissue and plant stress tolerance [41]. In this study, high-$CO_2$ treatments

yielded lower EL than air treatments (Figure 2), signifying that the high-$CO_2$ treatment induced minimal damage to the membrane of asparagus. $CO_2$ treatments decreased the electrolyte leakage in the MA packages, whereas no significant difference was found between the P packages. In the MA packages, ambient temperature (Air-24 h-20 °C and $CO_2$-24 h-20 °C) led to more severe damage than cold temperature (4 °C). High electrolyte leakage indicates serious damage to cellular tissue. Hypothetically, the effect of the high-$CO_2$ treatments on electrolyte leakage worked in terms of packaging film and storage temperature. High temperatures and improper gas environments in packages likely induce serious damage to asparagus.

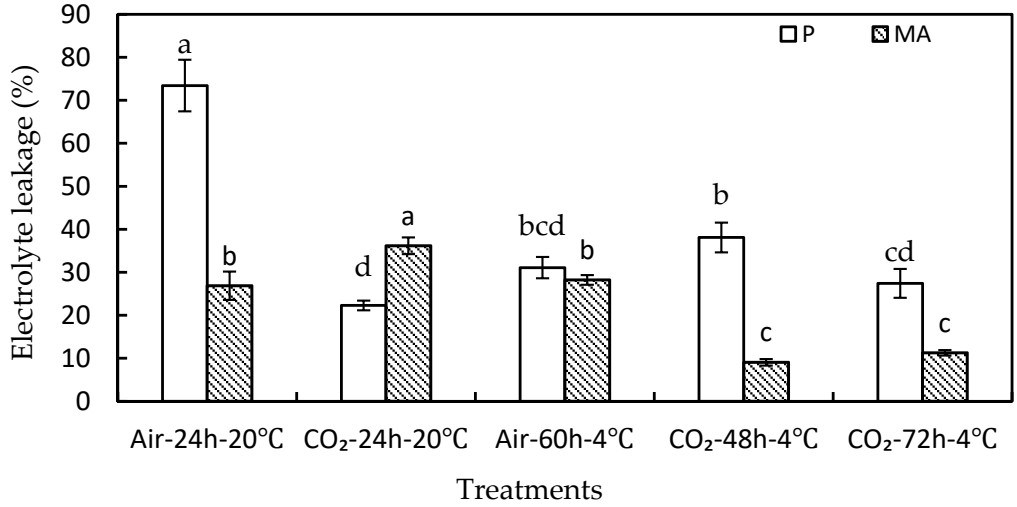

**Figure 2.** Electrolyte leakage of green asparagus treated with 60% $CO_2$ and air on the final storage day in P and MA packages. All groups were stored at 4 °C until the final day. Values are presented as means ± SE (*n* = 5). Different letters among the same packaging types indicate statistical difference using an LSD (Least Significant Difference) test at *p* < 0.05.

### 3.4. Microbiological Analysis

By modifying the gas environment, the growth of microorganisms can be inhibited to prolong shelf-life. $CO_2$, the main bacteriostatic component in the protective gas, was previously confirmed to inhibit the growth of most bacteria and molds. In this study, high-$CO_2$ treatments, regardless of the treatment temperatures and durations, yielded lower total aerobic bacteria, consistent with the fact that high $CO_2$ inhibits microbial growth and maintains aerobic respiration of fruits [42]. The $CO_2$-48 h-4 °C treatment yielded the lowest values of 4.11 log CFU·$g^{-1}$ and 4.31 log CFU·$g^{-1}$ in P and MA packages, respectively, on the final day (Table 4). In contrast, the Air-24 h-20 °C treatment yielded more aerobic bacteria, with values of 5.06 log CFU·$g^{-1}$ and 5.57 log CFU·$g^{-1}$ in P and MA packages, respectively. No yeast or mold were observed, with 0 log CFU·$g^{-1}$ in all treatments. In conclusion, 60% $CO_2$ for 48 h showed strong inhibition toward the growth of microorganisms, with lower total aerobic bacteria. The efficiency of inhibition by high $CO_2$ on microorganisms was also observed with other fruits and vegetables, such as table grapes (*Vitis vinifera* cv. Cardinal) and mango ("Carabao") [43,44]. The higher the $CO_2$ content, the greater the inhibition of microbial growth during the preservation of raw chicken meat in MA packages [45]. Hansen [46] reported that $CO_2$ inhibited the off-odor associated with spoilage and the growth of bacteria in fresh fish products, similar to the results of our research during asparagus postharvest storage. The exact mechanisms by which high $CO_2$ inhibits the growth of microorganisms are not clear, and many physiological processes may be involved, such as change in pH, disruption of microorganism cells, and changes in cell membranes and components [47].

**Table 4.** The growth of microorganisms (Total aerobic bacteria, yeast and mold) of green asparagus treated with 60% $CO_2$ on the initial and final storage days in P (the 25th day) and MA (the 30th day) packages.

| Treatments | Number of Microorganisms (log CFU·g$^{-1}$) | | | | | |
| --- | --- | --- | --- | --- | --- | --- |
| | Total Aerobic Bacteria | | | Yeast and Mold | | |
| | Initial | P | MA | Initial | P | MA |
| Air-24 h-20 °C | 4.42 a [z] | 5.06 b | 5.57 a | 0.00 a | 0.00 a | 0.00 a |
| $CO_2$-24 h-20 °C | 3.90 b | 4.43 c | 4.94 b | 0.00 a | 0.00 a | 0.00 a |
| Air-60 h-4 °C | 4.41 a | 5.35 a | 4.50 c | 0.00 a | 0.00 a | 0.00 a |
| $CO_2$-48 h-4 °C | 3.48 c | 4.11 d | 4.31 c | 0.00 a | 0.00 a | 0.00 a |
| $CO_2$-72 h-4 °C | 3.95 b | 4.29 c | 5.11 b | 0.00 a | 0.00 a | 0.00 a |

[z] Values are presented as means ± SE (*n* = 5). Different letters in the same column indicate statistically different means using an LSD (Least Significant Difference) test at *p* < 0.05.

### 3.5. Effect of High-$CO_2$ Treatments on Sensory Quality

The fresh weight-loss rate (%), visual quality, and off-odor of asparagus were evaluated to determine the effect of high-$CO_2$ treatment on sensory quality. In all treatments, MA packages (30 days) provided a shelf-life that was five days longer than the P packages (25 days). The loss of fresh weight increased over time (Figure 3), and the weight-loss rate in the P packages, with a range of 3.0–5.0%, was significantly higher than in MA packages (≤0.7%). On the final storage day, the highest fresh weight-loss rate was observed in Air-24 h-20 °C-P and $CO_2$-24 h-20 °C-P treatments, indicating that high temperature increased the fresh weight-loss rate. Irrespective of temperature and package type, $CO_2$ treatment slightly increased the fresh weight-loss rate, but no significant differences were evident within the same packaging film. $CO_2$ treatment reduced weight loss in fresh goji berries [48], indicating that high-$CO_2$ treatment prevents fresh weight loss.

Off-odor, which is due to volatile products such as methanol and acetaldehyde under anaerobic conditions [49], and asparagus rot were observed on the last storage day. High temperature led to the strongest off-odor in this study. $CO_2$-48 h-4 °C-MA and $CO_2$-72 h-4 °C-P treatments yielded the least off-odor, with scores of 1.7 and 2.3 (maximum = 5) on the 30th and 25th days, respectively (Figure 4). $CO_2$ treatment decreased off-odor both in the MA and P packages. Storage under moderate $O_2$ with $CO_2$ in MA packages (stabilizing near 10% $O_2$ + 9% $CO_2$) led to minor off-odor in baby spinach [49]. $CO_2$ could inhibit the growth of aerobic bacteria, yeast and molds [50], leading to decay and off-odor in asparagus. High $CO_2$ significantly alleviated strawberry decay during a period of cold storage [51].

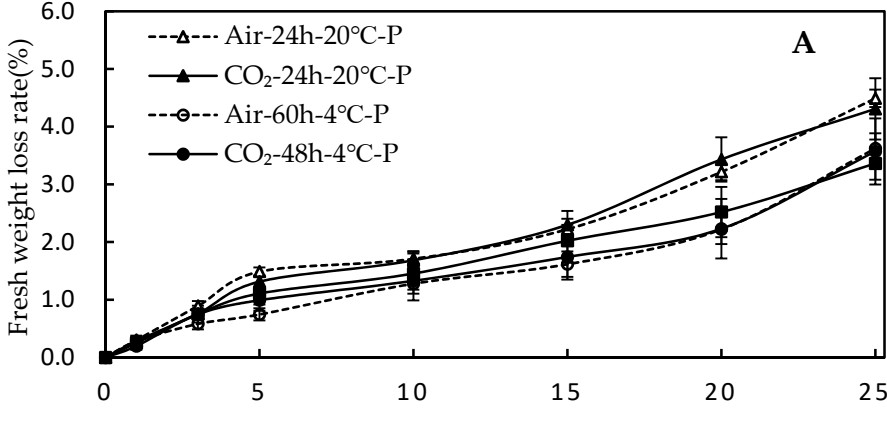

**Figure 3.** *Cont.*

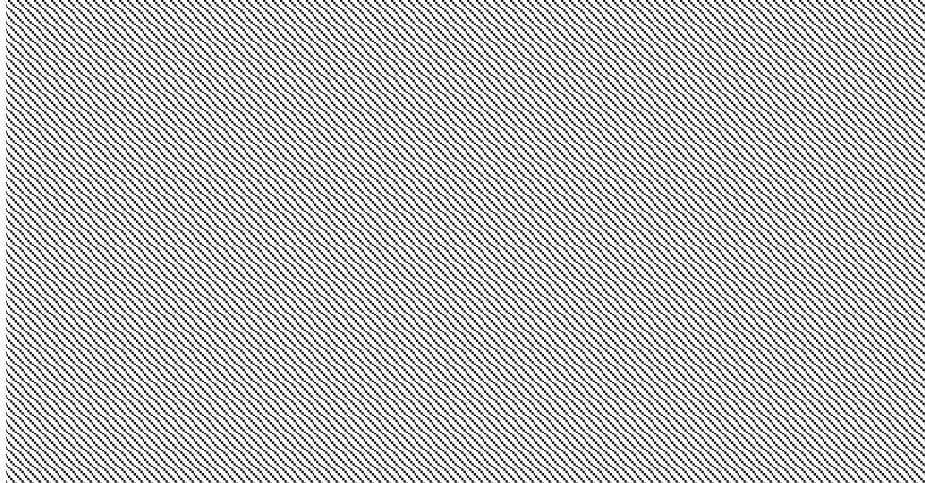

**Figure 3.** Changes in fresh weight-loss rate of green asparagus treatments with 60% $CO_2$ packaged with MA and P films. (**A**) Perforated packages (P); (**B**) 10,000 cc/m$^2$·day·atm oxygen transmission rate (OTR)-modified atmosphere packages (MA). All groups were stored at 4 °C until the final day. Values are presented as means ± SE (*n* = 5).

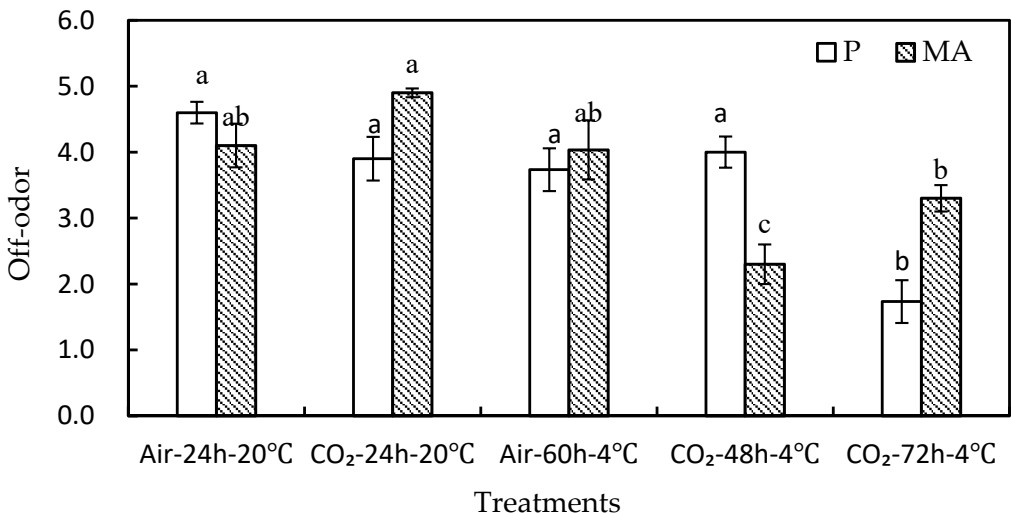

**Figure 4.** Off-odor of green asparagus in P and MA packages on the last storage day. Treatments of 60% $CO_2$ or air for 24 h at 20 °C and 60% $CO_2$ for 48 h, 72 h, or air treated for 60 h at 4 °C were conducted. All groups were stored at 4 °C until the final day. Values are presented as means ± SE (*n* = 5). Different letters marked among the same packaging types indicate statistical difference using an LSD (Least Significant Difference) test at *p* < 0.05.

Visual quality is the main consideration for consumers in purchasing asparagus. Characteristics of asparagus quality loss, such as longitudinal striation, bract opening, turgidity, base desiccation, color changes, and microorganism spoilage, were observed. By prolonging the storage time, the visual quality of asparagus decreased in all treatments (Figure 5). Air-60 h-4 °C-P and $CO_2$-48 h-4 °C-MA treatments maintained marketable quality, with scores higher than 3.0 after 25 days of storage. Treatments at 20 °C yielded the worst visual quality in both MA and P packages. High $CO_2$ treatments at 4 °C inhibited visual quality deterioration. The high-$CO_2$ treatment was effective in maintaining table grape (*V. vinifera* L. cv. Autumn Royal) quality, reducing strawberry water loss (*Fragaria vesca* L. cv. Mara de Bois), and inhibiting total decay [52–54].

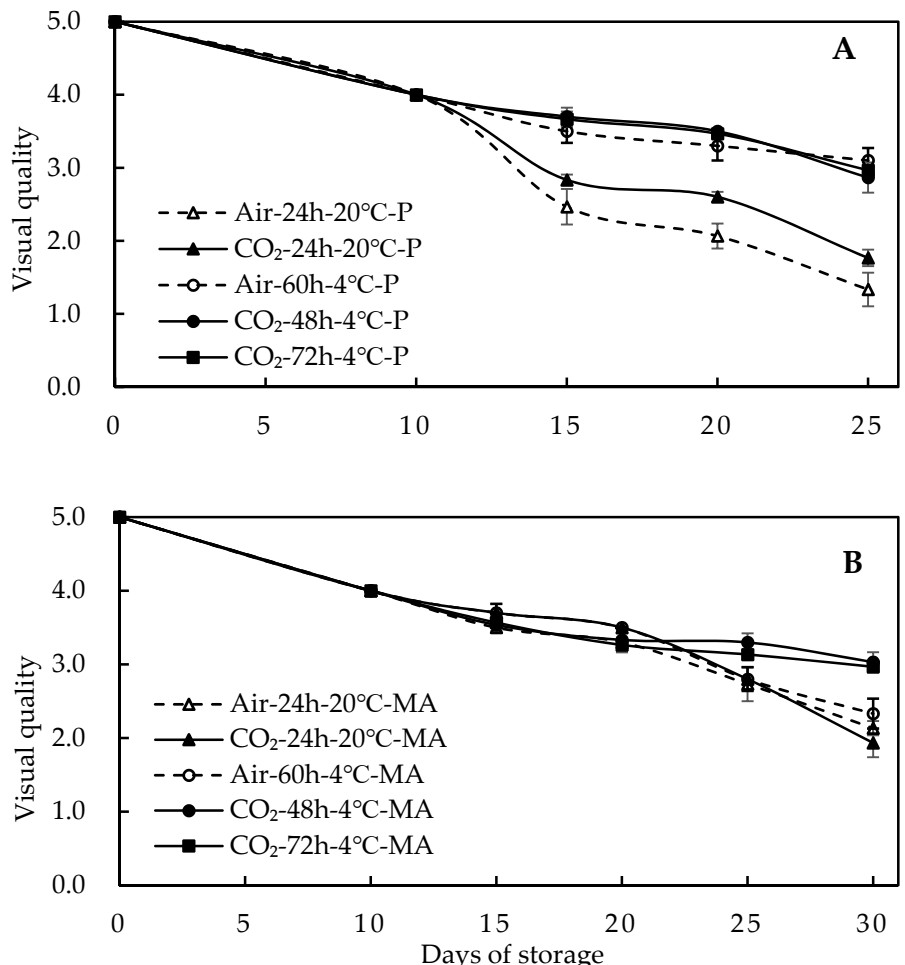

**Figure 5.** Changes in visual quality of green asparagus in P and MA packages. (**A**) Perforated packages (P); (**B**) 10,000 cc/m$^2$·day·atm OTR-modified atmosphere packages (MA). Treatments of 60% $CO_2$ or air pretreated for 24 h at 20 °C and 60% $CO_2$ treated for 48 h, 72 h or air for 60 h at 4 °C were conducted. All groups were stored at 4 °C until the final day. Values are presented as means ± SE (*n* = 5).

## 4. Conclusions

In this study, high-$CO_2$ treatment followed by MA packaging maintained shelf-life and quality characteristics of fresh asparagus. Color properties were best maintained with high-$CO_2$ treatments. A significant effect of high $CO_2$ treatment on asparagus color changes was induced by inhibiting the degradation of chlorophyll. The yellowing process was delayed due to the suppression of chlorophyllase activity, especially with the $CO_2$-48 h-4 °C-MA treatment. $CO_2$ at 60% for 48 h at 4 °C packaged with MA film ($CO_2$-48 h-4 °C-MA) was more favorable than the other treatments. With the $CO_2$-48 h-4 °C-MA treatment, high sensory quality, strong inhibition of microflora growth, higher chlorophyll content, and lower chlorophyllase activity were found. As a highly perishable horticultural crop, asparagus has a short shelf-life and is vulnerable to microorganisms and thrips. High-$CO_2$ treatment for a suitable duration provides the possibility of reducing the loss of quality during postharvest cold storage of asparagus.

**Author Contributions:** Conceptualization and methodology, L.-X.W., I.-L.C. and H.-M.K.; data curation, L.-X.W.; writing, L.-X.W.; writing—review and editing, H.-M.K. All authors have read and agreed to the published version of the manuscript.

**Funding:** This study was supported by the IPET through the Export Promotion Technology Development Program, with funding from the Ministry of Agriculture, Food and Rural Affairs (Nos. 114092-03 and 117035-03).

**Acknowledgments:** The authors would like to thank the laboratory members for their assistance.

**Conflicts of Interest:** The authors declare no conflict of interest.

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
