# Peer review of "Effect of High CO2 Treatment and MA Packaging on Sensory Quality and Physiological-Biochemical Characteristics of Green Asparagus (Asparagus officinalis L.) during Postharvest Storage"

_horticulturae, doi:10.3390/horticulturae6040084_

Round 1
Reviewer 1 Report
The English needs to be improved as not only does it it make the reading difficult, it also in some cases changes the meaning or raises the question of whether the authors are making correct statements.
for instance:
lines 147 and 148 " modified atmosphere packaging controls the gas environment" is incorrect modified atmosphere packaging as the name suggest modifies or influences the gas environment
line 151 atmosphere should be used not "air" as the concentration of oxygen in air does not change by 2%.
line 312 *CO2 treatment has been used to extend the shelf life of lettuce by inhibiting the bacterial growth for a long time," surely what is meant is that CO2 treatment has been used for many years to extend the shelf life of lettuce by inhibiting the bacterial growth,
These are just examples. These errors in the English do not give the reader confidence of the accuracy of certain statements which will be new to them.
There are also errors of the English such as;
The abstract should start " Green asparagus is vulnerable to thrips which carry microorganisms and cause the ........"
There are a few mistakes such as line 32 with the year, but these can be dealt with by further copy editing
The weight of the spears used does not seem to be given and should be given on line 83. Three spears seems a small sample size and should be explained as the typical quantity bought as a retail amount or if not, justified in some other respect.
At what temperature was the EL carried out (line 115) and what was the significance of the freezing and thawing out?
Where was the green asparagus sourced from?
As the texture is an important parameter, how does the time of season or the climatic conditions effect the spears? I do not see that this was considered. i assume that all the spears for all the treatments were sourced from the same location on the same day, but this is not stated and needs to be.
More information is needed on the five skilled panelists, such as what training, their age and gender distribution. Five seems a low number and in any work I have been involved in we have always had a minimum of eight.
Author Response
Thank you for your kind suggestions, our response was attached.
Please see the attachment

Reviewer 2 Report
The manuscript describes a traditional postharvest experiment. It is five replicates, but each replicate was of three spears. I am not familiar with differences between asparagus spears and would like to ask if three is enough. If one spear was responding differently than the others that make a big difference in the experiments. In Fig 1 the bars are denoted with letters, but it the figure legend is incomplete. It seems that the letters show differences between methods and within perforated or MA-packaged. That need to be explained. Same comment to Fig. 2 and 4. It is too many figures. The article could be shortened to be able to communicate the main message better.
Author Response
Thank you for your time and kind suggestions.
Please see the attachment

Round 2
Reviewer 1 Report
The paper reads much better and some of the data is better explained after your changes. I think it is now a clear and relevant paper, well done
Author Response
Thank you very much for your positive comments.